# A Novel Multi-Robot Task Allocation Model in Marine Plastics Cleaning Based on Replicator Dynamics

**Le Hong** [1,2,3], **Weicheng Cui** [2,3,*] and **Hao Chen** [1,2,3]

1   Zhejiang University-Westlake University Joint Training, Zhejiang University, Hangzhou 310024, China; hongle@westlake.edu.cn (L.H.); chenhao@westlake.edu.cn (H.C.)
2   Key Laboratory of Coastal Environment and Resources of Zhejiang Province (KLaCER), School of Engineering, Westlake University, Hangzhou 310024, China
3   Institute of Advanced Technology, Westlake Institute for Advanced Study, Hangzhou 310024, China
*   Correspondence: cuiweicheng@westlake.edu.cn

**Abstract:** As marine plastic pollution threatens the marine ecosystem seriously, the government needs to find an effective way to clean marine plastics. Due to the advantages of easy operation and high efficiency, autonomous underwater vehicles (AUVs) have been applied to clean marine plastics. As for the large-scale marine environment, the marine plastic cleaning task needs to be accomplished through the collaborative work of multiple AUVs. Assigning the cleaning task to each AUV reasonably and effectively has an essential impact on improving cleaning efficiency. The coordination of AUVs is subjected to harsh communication conditions. Therefore, to release the dependence on the underwater communications among AUVs, proposing a reliable multi-robot task allocation (MRTA) model is necessary. Inspired by the evolutionary game theory, this paper proposes a novel multi-robot task allocation (MRTA) model based on replicator dynamics for marine plastic cleaning. This novel model not only satisfies the minimization of the cost function, but also reaches a relatively stable state of the task allocation. A novel optimization algorithm, equilibrium optimizer (EO), is adopted as the optimizer. The simulation results validate the correctness of the results achieved by EO and the applicability of the proposed model. At last, several valuable conclusions are obtained from the simulations on the three different assumed AUVs.

**Keywords:** marine plastic pollution; autonomous underwater vehicles (AUVs); multi-robot task allocation (MRTA); replicator dynamics; equilibrium optimizer (EO)

## 1. Introduction

In the plan of "United Nations Decade of Ocean Science for Sustainable Development (2021–2030)", maintaining a healthy and clean marine environment received huge attention [1]. Moreover, there is a rising concern about the accumulation of floating plastic debris in the marine environment [2]. In 2018, China's survey of marine garbage showed that plastic debris accounted for 88.7% of floating waste on the sea, 77.5% of beach garbage, and 88.2% of submarine garbage [3]. As marine debris is difficult to degrade, its accumulation may lead to severe pollution and damage to the entire marine environment. It is particularly worth noting that almost half of the world's plastic waste comes from packaging [1]. Therefore, most plastics cannot be recycled or incinerated, causing massive destruction of resources and environmental pollution. It is also the primary source of marine plastic waste. Therefore, resolving and disposing of marine plastics is the key to maintaining a clean, sustainable, and productive ocean. The magnitude and the fate of cleaning marine plastics are still open questions [2]. The reduction of marine plastic waste is the shared responsibility of all countries globally, which is also the responsibility of every person. It is necessary to urge stakeholders from different governance levels, including government departments, public and private enterprises, civil organizations, and every individual in society, to reduce ocean plastic garbage and make bold innovations and actions. In recent

years, many countries have attached great importance to marine debris and microplastic pollution issues. Figuring out an effective way to clean marine plastics is the key to reduce the marine plastics pollution.

At first, manpower was mainly used to conduct various underwater tasks. However, utilizing trained divers to disposing marine plastics means that there would be many things to consider to ensure human safety and reach the demanded water depth to conduct cleaning work. On the other hand, till 2015, humans have produced at least 6.9 billion tons of plastic waste [4]. Moreover, due to improper human disposal, less than 9% of plastic waste is recycled, 12% is incinerated, and 79% is landfill or arbitrarily discarded into the environment. Therefore, replacing staffing with underwater tools is inevitable, especially under the rapid development of artificial intelligence. At the same time, underwater robots have been applied to various marine engineering fields, such as ship-hulls cleaning [5], ship-hulls inspection [6], deep ocean mining [7], etc. The same type AUVs could be applied in different scenarios when equipped with different tools. Especially for the underwater tracked vehicles (UTV), there are many different application scenarios for them. Such as, the UTV equipped with cutter bar for burring pipelines underwater [8], the UTV with rock-crushing tool for the underwater rock excavation [9], the UTV with ladder trench for the deep ocean mining [7], etc. Although there are a few AUVs for marine plastics cleaning, researchers still make great efforts to explore and innovate. In 2020, an Italian research team released a "lobster robot"-SILVER 2, which attracted the public's attention. The prominent positioning of this robot is to shoot and clean marine plastics under the sea. SILVER 2 can adapt well to various submarine topography, including seaweed, soil, rocks, sand, etc. It also shows good stability in the simulated water flow test [10].

With the gradual increase in the intelligence of AUVs and the complexity of underwater tasks, the performance of multiple AUVs is far better than that of a single AUV [11]. Therefore, utilizing and coordinating multi-robots to conduct large-scale cleaning tasks is necessary, which is also meaningful for the governance of marine plastic pollution. The key to successfully allocating tasks to multiple AUVs is coordinating the cleaning AUVs to obtain the optimal marine plastic-cleaning utility. At the same time, cooperation among a fleet of robotic agents is necessary and meaningful to improve the overall performance of any mission [12]. In terms of coordinating multi-robots to conduct a given task, it is illustrated by the name of multi-robots task allocation (MRTA). The robotic system is called multi-robots system (MRS). MRTA problem is a crucial concept in MRS. It can be modeled as two distinct sets: a set of tasks to be achieved and another set of robots capable of doing these tasks [13].

Due to the harsh underwater conditions, it is very difficult to control the single AUV work stably [14], let alone to connect AUVs on time and change their allocated tasks in time. Therefore, the MRTA problem for the AUVs needs to get the reliably pre-set task allocation values. There exist two main approaches to solve MRTA problems, which are market-based and optimization-based approaches. For the harsh environment, several allocation methods are designed to overcome it. For example, the location-aided task allocation framework method is specially designed to balance the objectives and the individual constraints of the AUVs [15]. Similar to the LAAF method, inspired by the evolutionary game theory (EGT), a specific and novel MRTA model for the AUVs is proposed on the basis of the optimization-based approaches. Benefits of the MRTA model combined with the replicator dynamics in the EGT can be illustrated as follows. First, in the process of MRTA, the demand of the whole multi-robot system and the allocated tasks of the robots are constantly and mutually adjusted and improved, which means many games are in progress [16]. They will also imitate and learn from the advanced experience of other agents to build their methodological system, similar to the biological evolution process [17]. Second, the EGT could emulate the bounded rationality of multi robots, which means the multi robots would evolve through mutual learning and adjustment and eventually form a strategic equilibrium and evolutionary stability [18]. In other words, most AUVs cannot consent to the optimal strategy of maximizing collective reward or minimizing the cost, and

the EGT can provide a relatively stable strategy after several rounds of dynamic evolution. Third, it has been proven that, under certain conditions, if the Karush–Kuhn–Tucker (KKT) first-order conditions of constrained optimization are satisfied, the equilibrium of EGT must exist [19]. At last, EGT has been successfully applied to the project about the several thorny problems and challenges; [20] modeled the demand-side management of a smart grid into a certain control networked evolutionary game, which minimize the total cost of the smart grid; [21] proposed a novel algorithm based on the evolutionary game theory, which addressed the challenges faced by dynamic placement of VMs successfully. All in all, the idea of applying several concepts of the EGT to the MRTA model is feasible and meaningful.

The contributions lie in three-folds. First, inspired the replicator dynamics of the EGT, a novel MRTA model is constructed on the basis of the optimization-based approach. It belongs to a continuous optimization problem, different from the discrete and combinatorial optimization formulation in the common MRTA model. Second, by introducing the replicator dynamics to the proposed MRTA model, the final task-allocation values would be in a relevantly stable state. Third, the EO algorithm is first applied to solve the MRTA problem. Through the simulation results, the EO algorithm can get the correct values when solving the proposed MRTA problem. Besides, several conclusions regarding the applicability of the proposed model and the impacts of the complete cleaning tasks are successfully obtained.

The paper is structured as follows. In Section 2, we introduce the related work involved in this paper, including the current situation of marine plastic pollution and the various solutions for the MRTA problems. Then, in Section 3, a novel MRTA model is constructed based on the optimization-based approach and replicator dynamics. Different from the common expression of the MRTA problem, the expression in this section belongs to the continuous optimization problem. Besides, after introducing the replicator dynamics to the proposed model, the expression includes utility function and stable function. To solve the proposed model, the EO algorithm is selected and illustrated in Section 4. Simulation results and discussions about the applicability of the proposed model, the impacts of the total tasks, and the effectiveness of the EO algorithm are shown in detail in Section 5. At last, conclusions are drawn in Section 6.

## 2. Preliminaries

### 2.1. Current Situation of Marine Plastic Pollution

The United Nations Environment Assembly listed marine plastic pollution as one of the major global environmental issues. Marine plastic pollution was also provided extremely high policy priorities, and governments were called for an urgent response [22]. This section presents the current deteriorating situation of marine plastic pollution from severity, pollution sources, and influence under the COVID-19.

Plastics in the marine environment receive more and more attention due to their persistence and impact on the ocean, wildlife, and even humans [23]. In April 2015, the Joint Group of Experts on the Scientific Aspects of Marine Environmental Protection (GESAMP) finished a report that concluded that marine plastic debris is as harmful to aquatic life as large marine plastic garbage. According to the relevant survey, marine biological species having marine plastic intake records has exceeded 800. This phenomenon has seriously affected the health and sustainable development of the marine ecosystem [24].

In different sea areas, the sources of marine plastic pollution are accordingly various. The garbage along the beach has long been subjected to sunlight, wave-washing, and biological effects. They are easily broken and decomposed into microplastics. Therefore, this type of process is considered one of the sources of marine plastics entering the sea. The surrounding near the estuary is often densely populated. A large amount of plastic waste from rivers would enter the waters. Therefore, this location is a hot spot for marine plastic pollution. The coastal areas are severely affected by human activities. Fisheries, aquaculture, shipping, sewage treatment plants, and so on are the sources of coastal plastics

entering the seas. Ocean circulation causes the global migration of marine debris. It has formed a so-called "garbage island" with a staggering area in the middle of the ocean. Although there are fewer human activities in the polar regions, the pollution of plastics in the polar seas is almost as severe as that in the coastal seas. This phenomenon shows that marine plastics have migrated to the polar regions and accumulated in the polar regions under ocean currents.

On the other hand, microplastics can be transported to the poles by the atmosphere, making the bars a gathering place for microplastics. Sampling in the deep sea and the abyssal zone is complex. Only a few countries have deep-sea sampling technologies. Chinese researchers discovered plastic and microplastics in the Mariana Trench, proving that plastic pollution has affected the deepest part of the seabed [25]. As COVID-19 is raging, a non-government organization (NGO) [26] stated that more than 1.5 billion masks had entered the ocean in 2020 [27]. Furthermore, the organization added that these masks would cost at least 450 years to degrade. At the same time, they are gradually being transformed into microplastics, which would have an extremely negative impact on marine life and the marine ecosystem.

From the above overviews, marine plastic pollution is a global problem that is inadequately managed. Therefore, addressing marine plastic pollution efficiently and effectively is the key for the countries to realize environmental protection and sustainable maritime development.

### 2.2. Algorithms for MRTA

As previously mentioned, there are two main approaches for MRTA problems, which are market-based approaches and optimization-based approaches. Market-based approaches are mainly inspired by the economic theory. They can provide an effective way to coordinate robots to perform relevant activities. The basis of market-based approaches is auctioning. They would consider both the robots' bidding and auction criteria (objective functions) while allocating tasks to robots [28]. However, they have many disadvantages: (1) The utilization of a central communication unit [29]; (2) scalability is ensured only for small and medium-sized problems [30]; (3) and the application of negotiation protocols. Compared with the market-based approaches, optimization-based techniques are more well-suited for distributed robots and generally produce near-optimal solutions more efficiently [31]. Besides, scalability is guaranteed even for a large number of robots. Research on using the optimization approaches to solve MRTA problems would be introduced as follows.

Two different solutions based on linear integer programming were proposed [32,33]. The same goal of them is to monitor a specific area and assign tasks to the robots. Mosteo and Montano used the traveling salesman problem (TSP) to formulate the MRTA problem and solved it through simulated annealing algorithm (SAA) [34]. Besides, SAA is combined with some heuristic algorithms to assign jobs to processors [33,35]. Similarly, a genetic algorithm (GA) is used to design a surveillance system that can track multiple targets and manage fires [36]. SAA and ant colony optimization (ACO) are combined to solve the path planning and MRTA problems [37]. Then, several MRTA solutions were proposed and tested extensively in some cases [38]. Robots are highly heterogeneous, and tasks are dynamic. Therefore, MRTA problems need more advanced approaches to be solved accurately. A distributed method is proposed to assign tasks to multiple robots based on particle swarm optimization (PSO) [39]. Authors of works [40,41] used swarm intelligence for task allocation in large-scale multiple robot systems. [42] dealt with the MRTA problem under spatial, temporal, and energetic constraints. The work in [43] proposed a solution to the MRTA problem using spatial queuing.

The meta-heuristic algorithm is proposed relative to the optimization algorithm. The optimization algorithm could obtain the optimal solution to a problem. However, the meta-heuristic algorithm is an algorithm based on intuition or experience. It can give a feasible solution to the problem at an acceptable cost, referring to the calculation time

and space. The degree of deviation between the viable solution and the optimal solution may not be predictable in advance. The work in [31,42] proposed two different MRTA problem solutions using different meta-heuristics, such as firefly algorithm, genetic quantum algorithm (GQA), artificial bee colony algorithm, and ant colony optimization (ACO). Equilibrium optimizer (EO) is a new meta-heuristic optimization algorithm proposed in 2020. It is inspired by the physical phenomenon of controlling volume mass balance. EO has the characteristics of solid optimization ability and fast convergence speed. The performance of EO has been verified by 58 mathematical functions, including unimodal, multimodal, mixed, and combined functions and three engineering benchmark problems. Due to the excellent performance of EO, the idea of utilizing EO to resolve the MRTA problem is proposed in this paper.

### 3. A Novel MRTA Model

Aiming to construct a MRTA model to get the optimal and stable allocated tasks, a distributed robotic system including n AUVs and a controlling center is assumed. Inspired by the evolutionary model and population dynamics, here, a novel MRTA model is constructed on the basis of the optimization-based approach, where the total weight of fouling material needed to be allocated is $w_{total}$ and the set of $[w_1, w_2, \ldots, w_n]$ is the weight of cleaning tasks allocated to the set of AUV-agents [AUV 1, AUV 2, $\ldots$, AUV n] by the controlling center. Therefore, the proposed MRTA model is different from the standard MRTA model with the combinational expression.

During the allocation, the sum of the agents in the set of $[w_1, w_2, \ldots, w_n]$ is required to be equal to $w_{total}$. The set of $[V_1(w_1), V_2(w_2), \ldots, V_n(w_n)]$ represent the goal function of the corresponding AUVs in the MRS after completing allocated tasks. In other words, they are the feedbacks that the related AUV agents give back to the controlling center after conducting tasks. Therefore, the total feedback of cleaning all the allocated plastic tasks in this distributed MRS is the sum of the set $[V_1(w_1), V_2(w_2), \ldots, V_n(w_n)]$. The relationship and interactions among n AUVs and the controlling center are shown in Figure 1.

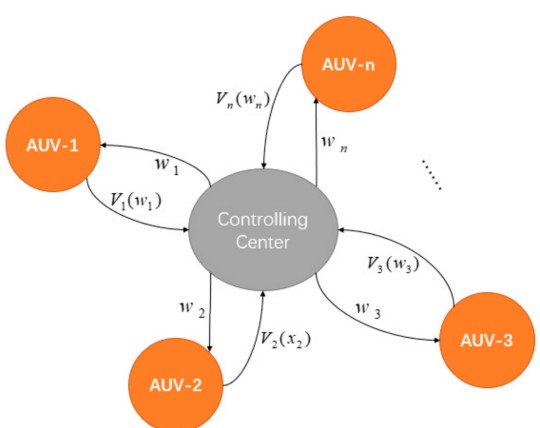

**Figure 1.** The architecture of the proposed distributed task-allocation system.

The components of the proposed distributed system are shown vividly in Figure 1. This distributed robotic system is composed of multiple distributed AUVs. Every distributed AUV has its goal function in cleaning marine plastics. There exists no master–slave relationship among these distributed AUVs, that is to say, they are equal during the task allocation of marine plastic cleaning. Moreover, they conduct the allocated cleaning tasks autonomously and individually. Preliminarily, the global constraint for this distributed task-allocation system is the total weight of the allocated tasks $[w_1, w_2, \ldots, w_n]$ must be $w_{total}$. The local constraint for this distributed-MRS is that the allocated task for the single AUV needs to satisfy the range $0 \leq w_i \leq w_{\mathrm{max}i}$. Under the above initial global constraint

and local constraints, this distributed task-allocation system can coordinate the AUVs to conduct the allocated tasks efficiently and harmoniously.

### 3.1. Fomulation of the MRTA Model

After sorting out the relationship between the multi-AUVs and the controlling center, we can first translate this kind of MRTA problem into a mathematical model based on the optimization solutions. Then, the MRTA model is translated and formed as follows:

$$
\min V(w) := \sum_{i=1}^{i=n} V_i(w_i)
$$

$$
s.t. \begin{cases} h_1(w) = \sum_{i=1}^{i=n} w_i - w_{total} = 0 \\ \forall i = 1, \ldots, n \quad w_{mini} \leq w_i \leq w_{\max i} \end{cases}
\tag{1}
$$

This model aims to minimize the goal function $V(w)$. To satisfy the initial requirements mentioned previously, this model is constructed under the global constraint that the sum of $w_i$(kg) must be $w_{total}$(kg). Each $w_i$ (kg) should be limited in the range from the minimum value to the maximum value of themselves, that is, $w_{mini} \leq w_i \leq w_{\max i}$.

### 3.2. Cost Function

The formulation of the cost function could be the most challenging part of an optimization-based MRTA problem. The general cost function of the MRTA model for underwater cleaning can be illustrated as follows:

$$
C = aw - cw
\tag{2}
$$

where $C[\$]$ denotes the cleaning costs of a single cleaning-AUV, and $w[kg]$ denotes the MPC tasks allocated to this cleaning-AUV. As we know, after the AUV finishes its allocated cleaning tasks, the owner of it would get profits from the government or the relevant companies. Therefore, during the task-allocation, it should consider the costs of AUVs and the profits of AUVs. The, then cleaning costs $a[\$/kg]$ per unit task and the cleaning profits $c[\$/kg]$ per unit tasks are chosen to be the other two coefficients in the general cost function. To be more specific, $a$ is the cost by the AUV when it finishes cleaning 1 kg plastics, including electricity cost, staffing consumption, etc., $c$ is the relevant department of the government affords $c$ to AUV's owners. That is to say, when the AUV is allocated 1 kg, the government or the relevant companies will pay $c$ to the AUV.

In this general cost function, the relationship between $C$ and $w$ is linear. Besides, $a$ and $c$ are always consistent. However, these two findings show that this linear expression is unpractical. There exist two main reasons. First, all relationships in real life are non-linear, and linear expression is just the simplification or the approximation of the fundamental engineering problems. On the other hand, there are many factors causing $a$ to change with $w$ changing. For example, due to the cleaning experience accumulated during the task conduction, when $w$ approaches its maximum cleaning weight $w_{\max}$, the changing rate of costs would decrease in real life. When $w$ approaches zero, the changing rate of cost should increase. The curve of the above two behaviors is shown in Figure 2.

Thus, we can see that changing rate of $C(w)$ is changeable and depends mainly on the value of $w$. Then, here, we utilize the logistic-type function to model the above phenomenon. The linear expression is successfully upgraded to the non-linear expression in Equation (3).

$$
C(w) = \frac{w}{r}\left(1 - \frac{w}{w_{\max}}\right)
$$
$$
s.t \quad w_{min} \leq w \leq w_{\max}
\tag{3}
$$

where we limit the value $w$(kg) to the range of $[w_{min}, w_{\max}]$. $C(w)$ still represents the total cost of the AUV, and $w_{\max}$ and $w_{\min}$ denote the maximum value and the minimum value of $w$. To reduce the number of coefficients in equation, $r$ is introduced as the only

coefficient in Equation (3). It denotes the plastics-cleaning ability of the AUV regardless of the value of $w$(kg), which would be calculated by the technical parameter of the AUV. Four main technical parameters of the cleaning AUVs are chosen to calculate $r$, which are the recharge mileage $m$[km], the maximum weight of the cleaning-plastics $w_{max}$[kg], the dive depth $d$[m], and the maximum speed $s$[km/h]. According to their importance in the marine plastics-cleaning, they are ranked in the order of $m$, $w_{max}$, $d$, and $s$. Then, $r$ can be calculated by the following Equation (4):

$$r = I_m \left( \frac{m}{(\sum_1^n m)/n} \right) + I_{w_{max}} \left( \frac{w_{max}}{(\sum_1^n w_{max})/n} \right) + I_d \left( \frac{d}{(\sum_1^n d)/n} \right) + I_s \left( \frac{s}{(\sum_1^n s)/n} \right) \quad (4)$$

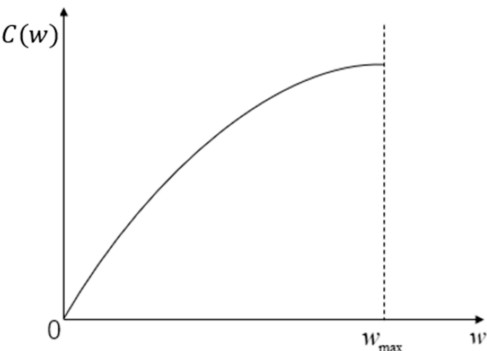

**Figure 2.** The curve of the upgraded non-linear model.

In Equation (4), the set of $[I_m, I_{w_{max}}, I_d, I_s]$ denotes the importance level of the corresponding index. $\frac{m}{(\sum_1^n m)/n}$, $\frac{w_{max}}{(\sum_1^n w_{max})/n}$, $\frac{d}{(\sum_1^n d)/n}$, and $\frac{s}{(\sum_1^n s)/n}$ represent the ratios of the current states to the average values. $n$ denotes the number of cleaning-AUVs. Based on the above construction and illustration of the cost function, the MRTA model can be further translated from Equation (1) to the following expression:

$$C(w) := \sum_{i=1}^{i=n} \frac{w_i}{r_i} \left( 1 - \frac{w_i}{w_{maxi}} \right)$$

$$s.t. \begin{cases} h_1(w) = \sum_{i=1}^{i=n} w_i - w_{total} = 0 \\ \forall i = 1, \ldots, n \quad 0 \le w_i \le w_{maxi} \end{cases} \quad (5)$$

### 3.3. Combination with Replicator Dynamics

As we have mentioned previously, the proposed MRTA model is inspired by the evolutionary game and population dynamics. The evolutionary game concepts are suitable for this MRTA model. Replicator dynamics is the core concept in evolutionary game theory, the definition of which is shown in definition 1 [18]. The replicator dynamics are introduced to the MRTA model to realize the stable allocated tasks among the AUVs. According to the definition of replicator dynamics and the corresponding Equation (10), a specific stable function for the proposed MRTA model is constructed and shown in Equation (6):

$$\frac{\partial w_i}{\partial t} = w_i \left( \frac{\partial C(w_i)}{\partial w_i} - \frac{\sum_{i=1}^n \left( \frac{\partial C(w_i)}{\partial w_i} \right)}{n} \right) \quad (6)$$

where $\frac{\partial w_i}{\partial t}$ reflects the changing rate of the $i$th AUV, which is assigned to $w_i$ cleaning tasks. Correspondingly, $\frac{\partial C(w_i)}{\partial w_i}$ represents the adaptability of the $i$th AUV and $\frac{\sum_{i=1}^n \left( \frac{\partial C(w_i)}{\partial w_i} \right)}{n}$ denotes the average adaptability among the $n$ AUVs. Therefore, if the value of $\frac{\partial w_i}{\partial t}$ is zero, it means that the $i$th AUV has stopped changing its allocated tasks. That is to say, in this

situation, the $i$th AUV has reached a relatively stable equilibrium state. In terms of the multiple AUVs, to reach a stable task-allocation, the modified replicator dynamics of all AUVs are needed to be zero or very close to zero. To solve the replicator dynamics by the optimization algorithm, Equation (6) is modified to Equation (7).

$$(w_i) = w_i \left| \frac{\partial C(w_i)}{\partial w_i} - \frac{\sum_{i=1}^{n} \left( \frac{\partial C(w_i)}{\partial w_i} \right)}{n} \right| \tag{7}$$

Based on the above work, a stable function for the proposed MRTA model is constructed as follows.

$$F(w) := \sum_{i=1}^{i=n} w_i \left| \frac{\partial C(w_i)}{\partial w_i} - \frac{\sum_{i=1}^{n} \left( \frac{\partial C(w_i)}{\partial w_i} \right)}{n} \right| \tag{8}$$

According to the mathematical expression of the stable function (8) based on the replicator dynamics, its goal is to minimize $F(w)$ to approach zero. As the goal of the cost function (5) and stable function (8) are both the minimize-optimization and positive-definite, the goal function of the proposed MRTA model is obtained by adding the two functions up, which is shown in Equation (9). After this kind of combination, the multi-objective optimization has also become single-objective optimization.

$$\min V(w) := C(w) + F(w)$$
$$s.t. \begin{cases} h_1(w) = \sum_{i=1}^{i=n} w_i - w_{total} = 0 \\ \forall i = 1, \ldots, n \; 0 \leq w_i \leq w_{\max i} \end{cases} \tag{9}$$

In the common optimization-based MRTA model, the objective function is just the utility function or the cost function. Therefore, compared to the common MRTA model, this novel model (9) not only satisfies the minimization of the cost, but also reaches a relatively stable state of the task allocation.

**Definition 1.** *Replicator Dynamic.*

The replicator dynamic is a dynamic differential equation, which describes the frequency of an adopted strategy in a specific population. It can be expressed by the following formula.

$$\frac{\partial x_i}{\partial t} = x_i [u(x_i, s_i) - u(x, x)] \tag{10}$$

In the above formula, $x_i$ is the proportion or probability of choosing pure strategy $s_i$ in a population, $u(x_i, x)$ represents the fitness when using pure strategy $s_i$, and $u(x, x)$ denotes the average fitness of the population.

## 4. EO Algorithm

The equilibrium optimization (EO) algorithm is first proposed by Faramarzi et al. EO originated from the control volume mass balance model, which is presented to estimate the dynamic and equilibrium states. For more information on the inspiration of EO, one may refer to [44]. Due to the advantages of simple principle, fast convergence, and easy implementation, EO has been widely applied to various optimization problems such as structural design optimization [45], economic dispatch [46], and image segmentation [47].

In EO, each individual with its concentration $\overrightarrow{C_o}$ is defined as a search agent, and each individual in the population is similar to a solution in the particle swarm optimization algorithm [48], $\overrightarrow{C_o}$ represents the position vector of an individual. Equilibrium optimization

algorithm consists of three parts, including initialization, equilibrium candidates and pool, concentration updating.

### 4.1. Initialization

As a meta-heuristic algorithm, the initial population is adopted in EO to start the search process. The concentrations of individuals are initialized by the following equation:

$$\overrightarrow{C}_{oi}^{ini} = \overrightarrow{C}_{omin} + r_i(\overrightarrow{C}_{omax} - \overrightarrow{C}_{omin}) \tag{11}$$

where $\overrightarrow{C}_{oi}^{ini}$ refers the initial concentration vector of the $i$th individual, $\overrightarrow{C}_{omax}$ and $\overrightarrow{C}_{omin}$ represent the lower limit vector and upper limit vector of the concentrations of the population, respectively, and $r_i$ indicates a random number in [0,1]. Then the individuals are evaluated by the fitness function.

### 4.2. Initialization Equilibrium Candidates and Pool

The equilibrium state represents the final convergence state of obtained solutions. In the initial optimization process, there is no knowledge about the final convergence state. Therefore, the equilibrium candidate $\overrightarrow{C}_{oe}$ is utilized to guide individuals in the population. In EO, equilibrium candidates include four best individuals based on their fitness values and an individual whose concentration is the mean of the four best individuals. The equilibrium pool is composed of the above five individuals.

$$\overrightarrow{C}_{oe,pool} = [\overrightarrow{C}_{oe(1)}, \overrightarrow{C}_{oe(2)}, \overrightarrow{C}_{oe(3)}, \overrightarrow{C}_{oe(4)}, \overrightarrow{C}_{oe(ave)}] \tag{12}$$

In each iteration, each individual updates its concentration by selecting one equilibrium candidate randomly from the equilibrium pool.

### 4.3. Concentration Updating

The individual's concentration updating is controlled by the exponential term $\overrightarrow{F}$.

$$\overrightarrow{F} = e^{-\overrightarrow{\lambda}(t-t_0)} \tag{13}$$

$$t = (1 - \frac{Cit}{Maxit})^{(a_2 \frac{Cit}{Maxit})} \tag{14}$$

where $Cit$ and $Maxit$ represent the current iteration and the maximal iteration, respectively. $t$ refers to the function of iterations which declines along with the increasing of iteration times. $a_2$ indicates a constant value that influences the exploitation ability of the equilibrium optimizer. $\lambda = [\lambda_1, \lambda_2, \ldots, \lambda_n]^T$ represents the random vector in the interval of [0,1], $t_0$ is calculated as follows:

$$t_0 = \frac{1}{\overrightarrow{\lambda}} \ln(-a_1 sign(r_0 - 0.5)[1 - e^{-\overrightarrow{\lambda}t}]) + t \tag{15}$$

where $a_1$ is a constant value which controls the exploration ability of EO, $sign(r_0 - 0.5)$ is employed to control the direction of exploitation and exploration, $\overrightarrow{r}_0$ indicates a random vector in the interval of [0,1].

The values of $a_1$ and $a_2$ are set to 2 and 1 respectively in this work. The final expression of the exponential term $\overrightarrow{F}$ can be calculated as follows:

$$\overrightarrow{F} = a_1 sign(\overrightarrow{r}_0 - 0.5)(e^{-\overrightarrow{\lambda}t} - 1) \tag{16}$$

Generation rate $\vec{G}$ plays an important role in EO. It is applied to enhance the exploitation ability of EO.

$$\vec{G} = \vec{G}_0 e^{-\vec{\kappa}(t - t_0)} \tag{17}$$

$$\vec{G}_0 = \vec{GCP}(\vec{C}_{oe} - \lambda\vec{C}_o) \tag{18}$$

$$\vec{GCP} = \begin{cases} 0.5r_1 & r_2 \geq GP \\ 0 & r_2 \leq GP \end{cases} \tag{19}$$

where $\vec{G}_0$ indicates the initial value. $\vec{GCP}$ is the generation rate control probability. $GP$ refers to the generation probability, which is set to 0.5 in this work. $r_1$ and $r_2$ represent two random numbers in the interval of [0,1]. $\vec{\kappa}$ refers to the decay vector. This study assumes $\vec{\kappa} = \vec{\lambda}$ according to the original EO algorithm. Therefore, the generation rate can be calculated as follows:

$$\vec{G} = \vec{G}_0\vec{F} \tag{20}$$

The final concentration updating formulation of EO is described as follows:

$$\vec{C}_o = \vec{C}_{oe} + (\vec{C}_o - \vec{C}_{oe})\vec{F} + \frac{\vec{G}}{\vec{\lambda V}}(1 - \vec{F}) \tag{21}$$

### 4.4. Pseudo Code of EO

Algorithm 1 is the pseudo code of the EO algorithm.

---

**Algorithm 1:** Equilibrium optimization algorithm.

```
 1  Initialize concentrations of individuals in the population;
 2  Initialize the parameters;
 3  Calculate the fitness value of each individual C_oi;
 4  Cit = 1;
 5  while Cit < Maxit do
 6      for i = 1 : N do
 7          Calculate the fitness value for the ith individual f(C⃗_oi);
 8          if f(C⃗_oi) < f(C⃗_oc(1)) then
 9              set C_oe(1) with C_oi and f(C⃗_oe(1)) with f(C⃗_oi);
10              else if f(C⃗_oi) > f(C⃗_oe(1)) & f(C⃗_oi) < f(C⃗_oe(2)) then
11                  set C_oc(2) with C_oi and f(C⃗_oe(2)) with f(C⃗_oi);
12                  else if f(C⃗_oi) > f(C⃗_oe(2)) & f(C⃗_oi) < f(C⃗_oe(3)) then
13                      set C_oe(3) with C_oi & f(C⃗_oe(3)) with f(C⃗_oi);
14                      else if f(C⃗_oi) > f(C⃗_oe(3)) and f(C⃗_oi) < f(C⃗_oc(1))
                          then
15                          set C_oe(4) with C_oi & f(C⃗_oe(4)) with f(C⃗_oi);
16                      end
17                  end
18              end
19          end
20      end
21      C⃗_oe(ave) = (C⃗_oe(1) + C⃗_oe(2) + C⃗_oe(3) + C⃗_oe(4))/4;
22      C⃗_oe,pool = {C⃗_oe(1), C⃗_oe(2), C⃗_oe(3), C⃗_oe(4), C⃗_oe(ave)};
23      Finish the memory saving;
24      Evaluate t by Eq.(14);
25      for i = 1 : N do
26          Choose a candidate from C⃗_oe,pool;
27          Generate vectors r⃗_0 and λ⃗_0 randomly;
28          Calculate F⃗ by Eq.(13);
29          Calculate GC⃗P by Eq.(19);
30          Calculate G⃗_0 by Eq.(18);
31          Calculate G⃗ by Eq.(17);
32      end
33      Cit ← Cit + 1;
34  end
```

---

*4.5. Initialization Constraint Handling*

Since the MPC-MRTA model is a constrained optimization problem, the penalty function method [49] is applied to handle the constraint in the MPC-MRTA model, and the constrained optimization model can be described as follows:

$$
\min V_c(w) := \sum_{i=1}^{i=n} V_i(w_i) + p_e |h_1(w)| \tag{22}
$$
$$
w.r.t.\ 0 \le w_i \le w_{\max i}
$$

where $P_e$ represents the penalty factor, which is set to 1000 in this work. If there is no constraint violation, $|h_1(w)|$ will be zero and positive otherwise. Under this conversion, the objective function now is $V_c(w)$ which serves as a fitness function in EO.

## 5. Simulation Results

*5.1. Parameters Setting of AUVs*

To make the proposed MRTA model applied to the real multi AUV systems, the parameters of the assumed AUVs refer to the two types of Hebao DF-H4 water-surface cleaning-AUVs. The four main technical parameters related to the cleaning of the marine plastic are selected, which are the recharge mileage $m$ [km], the maximum weight of the cleaning-plastics $w_{max}$ [kg], the dive depth $d$ [m], and the maximum speed $s_{max}$ [km/h]. Three types of AUVs for marine plastics cleaning are assumed. According to their sizes, the abbreviation of the three assumed AUVs are S-AUV, M-AUV, and L-AUV. The four main technical parameters of the three AUVs are shown in Table 1.

**Table 1.** Four main technical parameters of the three types assumed AUVs.

|       | $m$ [km] | $w_{max}$ [kg] | $d$ [m] | $s_{max}$ [km/h] | $r$  |
|-------|----------|----------------|---------|------------------|------|
| S-AUV | 48       | 12             | 380     | 32               | 0.93 |
| M-AUV | 60       | 16             | 270     | 24               | 1.07 |
| L-AUV | 75       | 20             | 220     | 16               | 1.09 |
| $I$   | 0.4      | 0.3            | 0.2     | 0.1              | N/A  |

Table 1 shows the four main technical parameters of the S-AUV, and M-AUV and L-AUV present the important factor $I$ of the four main parameters acting in cleaning marine plastics. Therefore, we assume the critical factor of the four parameters, which are $I_m$, $I_{w_{max}}$, $I_d$, $I_s$, are equal to 0.4, 0.3, 0.2, 0.1, respectively. Then, the cleaning ability factor $r$ of the three AUVs can be calculated by Equation (4). Moreover, the exact values of $r$ are also shown in Table 1. The rank of AUVs about the cleaning ability $r$ is L-AUV, M-AUV, and S-AUV.

*5.2. Applicability of the System Model*

To explore the applicability of the proposed MRTA model, three scales of the multi-robots system are selected to simulate: the small-scale system, middle-scale system, and the large-scale system. The composition of the three systems is shown in Table 2, which shows the detailed number of the three types of AUVs and the total number of AUVs. In Table 2, $i$ denotes the serial number of the AUV, and n represents the total number of AUVs. AUV is known to be expensive, bulky, and complicated. Therefore, three AUVs including 1 S-AUV, 1 M-AUV, and 1 L-AUV are enough for a small-scale system. The serial number of the three AUVs is 1, 2, 3, respectively. Based on the small-scale system, in the middle-scale system, the total number of AUVs is increased to 6. Correspondingly, the numbers of the three types of AUVs are all increased to 2. The serial numbers of the composed S-AUV, M-AUV, and L-AUV are $i = 1, 4$, $i = 2, 5$, and $i = 3, 6$, respectively. The large-scale system is composed of 12 AUVs, which includes 4 S-AUVs, 4 M-AUVs, and 4 L-AUVs. The corresponding serial number of them are $i = 1, 4, 7, 10$, $i = 2, 5, 8, 11$, $i = 3, 6, 9, 12$.

**Table 2.** Composition of the three scales of the multi-AUVs system.

| System Scale | S-AUV | M-AUV | L-AUV | $n$ |
|---|---|---|---|---|
| Small-Scale | $1(i=1)$ | $1(i=1)$ | $1(i=1)$ | 3 |
| Middle-Scale | $2(i=1,4)$ | $2(i=2,5)$ | $2(i=3,6)$ | 6 |
| Large-Scale | $4(i=1,4,7,10)$ | $4(i=2,5,8,11)$ | $4(i=3,6,9,12)$ | 12 |

Initially, considering the common loads of the AUV and the estimated cleaning need of the marine plastics, the total marine plastics cleaning tasks $w_{total}$ is set to 18 kg and $w_{min}$ for all AUVs is set to 1 kg. To compare the simulation performance under the widespread algorithms, simulations would be conducted not only by the EO algorithm but also by particle swarm optimization (PSO) algorithm and the genetic algorithm (GA). During the simulations under the PSO, the inertia weight is set to be 1 and the two acceleration constants are set to be 1.5 and 2.0, respectively. As for the simulations under the GA, the mutation probability is set to be 0.5 and the crossover probability is set to be 0.8. By repeating the simulation program 100 times and taking the average value, simulation results in the three constructed systems are shown in Tables 3–5.

**Table 3.** Allocated tasks for the AUVs and the value of the goal function in the small-scale system.

| | $w_1$ | $w_2$ | $w_3$ | $V$ |
|---|---|---|---|---|
| EO | 4.66 | 5.95 | 7.39 | 10.8323 |
| PSO | 4.73 | 6.01 | 7.26 | 10.8963 |
| GA | 4.66 | 5.95 | 7.39 | 10.8324 |

**Table 4.** Allocated tasks for the AUVs in the middle-scale system.

| | $w_1$ | $w_2$ | $w_3$ | $w_4$ | $w_5$ | $w_6$ | $V$ |
|---|---|---|---|---|---|---|---|
| EO | 2.66 | 2.87 | 3.47 | 2.66 | 2.87 | 3.47 | 14.1164 |
| PSO | 2.66 | 2.87 | 3.47 | 2.66 | 2.87 | 3.47 | 14.1169 |
| GA | 2.70 | 2.92 | 3.38 | 2.70 | 2.92 | 3.36 | 14.1790 |

**Table 5.** Allocated tasks for the AUVs in the large-scale system.

| | $w_1$ | $w_2$ | $w_3$ | $w_4$ | $w_5$ | $w_6$ | $w_7$ | $w_8$ | $w_9$ | $w_{10}$ | $w_{11}$ | $w_{12}$ | $V$ |
|---|---|---|---|---|---|---|---|---|---|---|---|---|---|
| EO | 1.65 | 1.34 | 1.51 | 1.65 | 1.34 | 1.50 | 1.65 | 1.34 | 1.51 | 1.65 | 1.34 | 1.51 | 15.8337 |
| PSO | 1.65 | 1.33 | 1.51 | 1.65 | 1.33 | 1.51 | 1.65 | 1.33 | 1.51 | 1.65 | 1.33 | 1.51 | 15.8404 |
| GA | 1.65 | 1.33 | 1.51 | 1.65 | 1.33 | 1.51 | 1.65 | 1.33 | 1.51 | 1.65 | 1.33 | 1.51 | 15.8343 |

By analyzing the simulation results, the allocated tasks are all subjected to the constraints in the system model (10), and the goal function is well achieved. This phenomenon indicates that the system model could be applied to different systems with different scales. Analyzing the results simulated by the three algorithms, it can be concluded that although the EO algorithm performs slightly better, there is no significant difference in the values they obtained. Due to the popularity of the PSO and GA, the results can also indicate that the EO algorithm can get the correct and expected values.

*5.3. Impacts of the Total Tasks*

The previous simulations are all conducted when $w_{total}$ is set to 18 kg. To figure out the impacts of the $w_{total}$ on the values of allocated tasks and the goal function, four more simulations are conducted in the constructed small-scale system and middle-scale system under $w_{total} = 9$ kg and $w_{total} = 36$ kg. To satisfy the constraint that $w_i$ needs to be above the $w_{min}$, two more simulations in the large-scale system are conducted under $w_{total} = 36$ kg and $w_{total} = 72$ kg. The other simulation settings are unchanged. The simulation results are shown in Tables 6–11. Like the former simulations, the difference

between the results achieved by the three algorithms is almost non-existent. Therefore, the values under the EO algorithm shown in these tables are selected for further comparison.

**Table 6.** Allocated tasks for the AUVs and the value of the goal function in the small-scale system under $w_{total} = 9$ kg.

|     | $w_1$ | $w_2$ | $w_3$ | $V$ |
| --- | --- | --- | --- | --- |
| EO  | 2.66 | 2.87 | 3.47 | 7.0579 |
| PSO | 2.66 | 2.87 | 3.47 | 7.0585 |
| GA  | 2.66 | 2.87 | 3.47 | 7.0585 |

**Table 7.** Allocated tasks for the AUVs and the value of the goal function in the small-scale system under $w_{total} = 36$ kg.

|     | $w_1$ | $w_2$ | $w_3$ | $V$ |
| --- | --- | --- | --- | --- |
| EO  | 8.67 | 12.10 | 15.22 | 8.6756 |
| PSO | 8.67 | 12.10 | 15.22 | 8.6759 |
| GA  | 8.67 | 12.10 | 15.22 | 8.6759 |

**Table 8.** Allocated tasks for the AUVs and the value of the goal function in the middle-scale system under $w_{total} = 9$ kg.

|     | $w_1$ | $w_2$ | $w_3$ | $w_4$ | $w_5$ | $w_6$ | $V$ |
| --- | --- | --- | --- | --- | --- | --- | --- |
| EO  | 1.65 | 1.33 | 1.51 | 1.65 | 1.33 | 1.51 | 7.9162 |
| PSO | 1.65 | 1.33 | 1.51 | 1.65 | 1.33 | 1.51 | 7.9169 |
| GA  | 1.65 | 1.33 | 1.51 | 1.65 | 1.33 | 1.51 | 7.9169 |

**Table 9.** Allocated tasks for the AUVs and the value of the goal function in the middle-scale system under $w_{total} = 36$ kg.

|     | $w_1$ | $w_2$ | $w_3$ | $w_4$ | $w_5$ | $w_6$ | $V$ |
| --- | --- | --- | --- | --- | --- | --- | --- |
| EO  | 4.66 | 5.95 | 7.39 | 4.66 | 5.95 | 7.39 | 21.6647 |
| PSO | 4.97 | 6.07 | 6.93 | 4.97 | 6.06 | 7.00 | 22.0600 |
| GA  | 5.27 | 6.17 | 6.74 | 5.27 | 5.81 | 6.74 | 22.4686 |

**Table 10.** Allocated tasks for the AUVs and the value of the goal function in the large-scale system under $w_{total} = 36$ kg.

|     | $w_1$ | $w_2$ | $w_3$ | $w_4$ | $w_5$ | $w_6$ | $w_7$ | $w_8$ | $w_9$ | $w_{10}$ | $w_{11}$ | $w_{12}$ | $V$ |
| --- | --- | --- | --- | --- | --- | --- | --- | --- | --- | --- | --- | --- | --- |
| EO  | 2.66 | 2.87 | 3.47 | 2.66 | 2.87 | 3.47 | 2.66 | 2.87 | 3.47 | 2.66 | 2.87 | 3.47 | 28.2339 |
| PSO | 2.66 | 2.88 | 3.47 | 2.67 | 2.89 | 3.40 | 2.67 | 2.88 | 3.48 | 2.67 | 2.88 | 3.45 | 28.2616 |
| GA  | 2.77 | 2.90 | 3.38 | 2.77 | 2.89 | 3.33 | 2.75 | 2.96 | 3.31 | 2.73 | 2.97 | 3.22 | 28.5068 |

**Table 11.** Allocated tasks for the AUVs and the value of the goal function in the large-scale system under $w_{total} = 72$ kg.

|     | $w_1$ | $w_2$ | $w_3$ | $w_4$ | $w_5$ | $w_6$ | $w_7$ | $w_8$ | $w_9$ | $w_{10}$ | $w_{11}$ | $w_{12}$ | $V$ |
| --- | --- | --- | --- | --- | --- | --- | --- | --- | --- | --- | --- | --- | --- |
| EO  | 4.87 | 6.01 | 7.25 | 4.87 | 5.91 | 7.12 | 4.87 | 6.01 | 7.05 | 4.87 | 5.91 | 7.25 | 43.7385 |
| PSO | 5.29 | 6.08 | 6.64 | 5.30 | 6.02 | 6.98 | 5.28 | 5.61 | 6.71 | 5.26 | 6.06 | 6.78 | 44.7677 |
| GA  | 5.42 | 5.97 | 6.55 | 5.48 | 5.47 | 6.42 | 5.54 | 6.10 | 7.03 | 5.47 | 5.86 | 6.69 | 45.4573 |

Therefore, in Figures 3–5, the selected values achieved by the EO algorithm are well presented by the line charts. When the simulations are conducted in the small-scale system, $w_1$, $w_2$, $w_3$ are the total allocated tasks for the S-AUV, M-AUV, and L-AUV, respectively. When the middle-scale system and the large-scale system are chosen to be simulated, the total allocated tasks for the three types of AUVs are the sum of the allocated tasks for the corresponding AUVs, for which serial numbers have been set previously in Section 5.2.

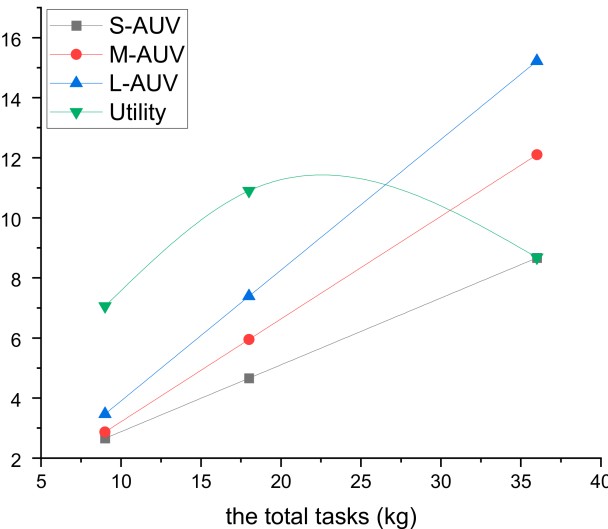

**Figure 3.** The line chart of the values simulated in the small-scale system under three different $w_{total}$.

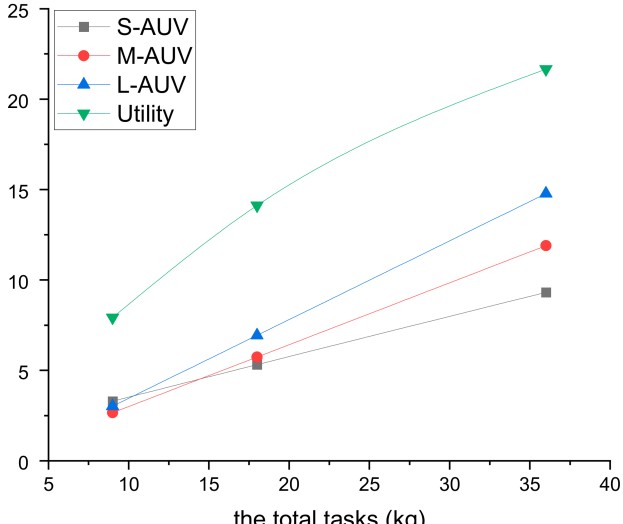

**Figure 4.** The line chart of the values simulated in the middle-scale system under three different $w_{total}$.

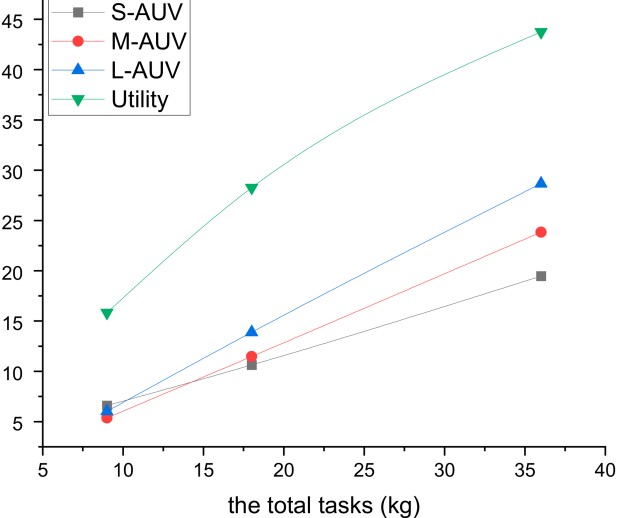

**Figure 5.** The line chart of the values simulated in the large-scale system under three different $w_{total}$.

To analyze the charts comprehensively, we can find that with $w_{total}$ increases, the changing rate of the allocated tasks for the L-AUVs would be the biggest among the tasks allocated for the three types of AUVs. Similarly, the increase rate of the allocated tasks for the M-AUVs is larger than that of the S-AUVs. However, from Table 8, we can find that L-AUV is not allocated the most tasks all the time, which reflects that the relationship between the cleaning-ability and the allocated tasks is non-linear.

Generally, the changing rate of L-AUVs is the largest, followed by the M-AUVs. The cause of this phenomenon lies in the fact that the rank of changing-ability factor *r* is: L-AUV, M-AUV, and S-AUV. However, comparing Table 8 with the other tables, it needs to be noted that the AUV with larger cleaning ability would not be allocated more tasks under any circumstance, which reflects the non-linear relationship between the AUV's cleaning ability and the allocated tasks. Besides, the goal function V in the small-scale system decreases when $w_{total}$ increases from 18 kg to 36 kg, which indicates that V is not proportional to the $w_{total}$, either.

To sum it up, through the three algorithms' simulation results, compared to the two other popular algorithms, the EO algorithm is validated to achieve a similar task-allocation value with a slightly better performance. This phenomenon proves that the EO algorithm could achieve the correct and expected values for the proposed MRTA model. By assuming and listing three systems with different numbers of AUVs, all the final task-allocation values satisfy the constraints in the proposed model, which illustrates that the applicability of the proposed MRTA model is far-ranging. Besides, when the $w_{total}$ is increased, changing rate of the allocated tasks for the AUVs with the better changing-ability factor r would be larger.

## 6. Conclusions

Owing to the harsh communication conditions for the AUVs, inspired by the evolutionary game theory and population dynamics, a novel and specific MRTA model for marine plastics cleaning is established. To get the optimized and relatively stable task-allocation values, the goal function of this novel MRTA model consists a cost function and replicator dynamics of AUVs. Then, to solve this MRTA problem, the EO algorithm is chosen for its better performance in the other fields. It is the first time that the EO algorithm has been applied to the MRTA problem. Through the simulations, the following conclusions could be obtained. First, the EO algorithm is verified for being able to calculate the correct and expected values in the proposed MRTA model. Second, the proposed MRTA model is applicable for the different scales of the multi-robot system. Finally, the AUV with a larger cleaning ability factor r, determined by its four main technical parameters, would be allocated more tasks and increase faster with the total tasks increase.

It must be admitted that to apply the replicator dynamics to the MRTA model, the constraints and the goal function in the model are set to be relatively simple. Compared with the existing literatures, the difficulties of this research are as follows. First, there are no powerful underwater robots designed for cleaning marine plastics in the market. Second, to make this research practical, there will be a long way for us to study. Future work could include constructing a more complex MRS for underwater operations with more constraints and combining the proposed MRTA model with multi-robots path planning problem.

**Author Contributions:** Conceptualization, L.H. and H.C.; methodology, L.H. and W.C.; software, H.C.; validation, L.H. and H.C.; formal analysis, L.H. and H.C.; investigation, W.C.; resources, W.C.; writing-original draft preparation, L.H.; funding acquisition, W.C.; project administration, W.C. All authors have read and agreed to the published version of the manuscript.

**Funding:** The author(s) disclosed receipt of the following financial support for the research, authorship, and/or publication of this article: This work was supported by Zhejiang Key R&D Program No.2021C03157, the "Construction of a Leading Innovation Team" project by the Hangzhou Municipal government, the Startup funding of New-joined PI of Westlake University with grant number (041030150118) and the funding support from the Westlake University and Bright Dream Joint Institute for Intelligent Robotics.

**Conflicts of Interest:** The authors declare no conflict of interest. The funders had no role in the design of the study, in the collection, analyses, or interpretation of data, in the writing of the manuscript, or in the decision to publish the results.

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
