# Peer review of "A Novel Multi-Robot Task Allocation Model in Marine Plastics Cleaning Based on Replicator Dynamics"

_jmse, doi:10.3390/jmse9080879_

Round 1

Reviewer 1 Report

It is my belief, that recent updates to the paper made it suitable for publication. The overall goal of the paper is better justified, although the results should still be treated as preliminary.

Non-critical issues.

  • Reference [10] does not link directly to content implied in the text,.
  • Figures 3-5 diminish the overall quality of the paper (maybe something like 'pgfplots' could be used here).
  • Visual artifact in Table 11 (probably just an editorial issue).
  • In (8) the sigma sign could be smaller.
  • In (5), (8), you want to express that respective functions are to be minimized, but the notation suggests that the minimal value is given therein.

Author Response

Thank you very much for your careful reading and helpful comments on our manuscript entitled “A Novel Multi-Robot Task Allocation Model in Marine Plastics Cleaning Based on Replicator Dynamics” (ID: jmse-1322674). These valuable comments have been very helpful for revising and improving our paper. We have studied those comments carefully and have made corrections which we hope will meet with your approval.

The main corrections in the paper and the responses to the editor and reviewer are described as follows. The comments are presented in italic, and our response is highlighted in grey.

All the changes are shown in red color.

 Reviewer #1:

Comment No. 1:

It is my belief, that recent updates to the paper made it suitable for publication. The overall goal of the paper is better justified, although the results should still be treated as preliminary. Reference [10] does not link directly to content implied in the text.

Response to comment No. 1:

Many thanks for your careful review. After checking the literature [10] and its relations to the paper, we finally deleted the citation of it. The modification of it is shown in:

Page 2, Line 73-74. One sentence has been deleted.

Comment No. 2:

Figures 3-5 diminish the overall quality of the paper (maybe something like 'pgfplots' could be used here).

Response to comment No. 2:

Thanks for your helpful suggestion. We have improved the quality of figures 3-5 by ’Origin’. The modifications are shown in:

Page 15-16, Figure 3-5.

Thanks again for your careful suggestions. 

Comment No. 3:

Visual artifact in Table 11 (probably just an editorial issue).

Response to comment No. 3:

Thanks for your careful review. In order to avoid the visual artifact, after modifying the paper, we checked it in the unmarked mode to ensure that there are no errors in the final issue.

Comment No. 4:

In (8) the sigma sign could be smaller. In (5), (8), you want to express that respective functions are to be minimized, but the notation suggests that the minimal value is given therein.

Response to comment No. 4:

Thanks for your helpful suggestions. These two mistakes have been modified in:

Page 8, Equation (5).

Page 8, Equation (8).

Thanks again for your careful review.

Reviewer 2 Report

The topic is timely; hence it aims at the readership of the journal. The authors have discussed the proposed approach thoroughly. However, I believe that there is still room for improvement in the following way:

1/ Benefit of the proposed method is not clear. This should be clarified. The objective/problem statement needs to be explained. What are the novelties of the proposed method? what the research challenges/motivations of the paper are?

2/ The quality of the introduction can be improved. For example, some new works related to underwater vehicles and new applications of underwater robots. To help the authors in this direction, I suggest the following reference: https://doi.org/10.3390/s21030747, https://doi.org/10.1016/j.apor.2019.02.019, https://doi.org/10.1016/j.apor.2017.12.005, https://doi.org/10.1177/1475090217735934

3/ Please highlight which section(s) discusses each of the contributions. This way there will be cohesiveness in the manuscript contents.

4/ The contributions of this article should be improved in comparison with the existing literature. Compared with existing results, the advantages of the paper should be further highlighted and the occurred difficulties of conducted topic can be explained.

5/ In the simulation part, more design parameters are recommended. A detailed discussion of the figures is helpful to illustrate the results. It is better to explain how the values of the parameters in the proposed method are adjusted?

6/ To confirm the effectiveness and practicality of the proposed approach, an experiment is strongly suggested since the theoretical approach of this study seems ordinary.

7/ The readability and presentation of the work should be improved. Some grammar issues and typos need to be corrected.

Author Response

Reviewer #2:
Comment No. 5:

The topic is timely; hence it aims at the readership of the journal. The authors have
discussed the proposed approach thoroughly. However, I believe that there is still room
for improvement in the following way: Benefit of the proposed method is not clear. This
should be clarified. The objective/problem statement needs to be explained. What are
the novelties of the proposed method? what the research challenges/motivations of the
paper are?

Response to comment No. 5:

Many thanks for your critical comments. Benefits of the proposed method include two
parts. The first part are the benefits of the proposed MRTA model inspired by the
replicator dynamics in the evolutionary game theory. We have added relevant
introduction about this kind of benefits in:

Page 3, Line 100. Benefits of the MRTA model combined with the replicator dynamics
in the EGT can be illustrated as follows.

The corresponding four benefits of the proposed MRTA model follow this modification.
The second part of the proposed method are the overall benefits of the method, which
have been shown clearly in Page 3, Line 118-128.

This paper has two main motivations, which are the marine plastics pollution and the
intermittent and unreliable underwater communications. To make these two
motivations clearer, the modifications are shown in:

Page 2, Line 50. Figuring out an effective way to clean marine plastics is the key to
reduce the marine plastics pollution.

Page 2, Line 88. Due to the harsh underwater conditions it is very difficult to control
the single AUV work stably [17],

After these modifications, this paper becomes more attractive. Thanks again for your
careful review.

Comment No. 6:

The quality of the introduction can be improved. For example, some new works related
to underwater vehicles and new applications of underwater robots. To help the authors
in this direction, I suggest the following reference:
https://doi.org/10.3390/s21030747,
https://doi.org/10.1016/j.apor.2019.02.019
,
https://doi.org/10.1016/j.apor.2017.12.005
,
https://doi.org/10.1177/1475090217735934

Response to comment No. 6:

Thanks for your suggestions. After reading your suggested four papers, the first paper
is related to the allocation algorithm for a single-AUV under the harsh underwater
environment. Therefore, we decided to cite this paper in:

Page 2, Line 89. it is very difficult to control the single AUV work stably [17],

As for the other three papers, they studied the underwater tracked vehicles equipped
with different tools in different application scenarios. We cited them in:

Page 2, Line 63-67. The same type AUVs could be applied in different scenarios when
equipped with different tools. Especially for the underwater tracked vehicles (UTV),
there are many different application scenarios for them. Such as, the UTV equipped
with cutter bar for burring pipelines underwater [10], the UTV with rock crushing tool
for the underwater rock excavation [11], the UTV with hadal trench for the deep ocean
mining [9], etc..

Many thanks for your helpful suggestions and the four suggested papers. After adding
the citations of these four helpful papers, the quality of the introduction has been
improved a lot.

Comment No. 7:

Please highlight which section(s) discusses each of the contributions. This way there
will be cohesiveness in the manuscript contents.

Response to comment No. 7:

Thanks for your helpful comments. The detailed structural flow is modified in:

Page 3, Line 131-142. The paper is structured as follows. In Section 2, we introduce the related
work involved in this paper, including the current situation of marine plastic pollution and the
various solutions for the MRTA problems. Then, in Section 3, a novel MRTA model is
constructed based on the optimization-based approach and replicator dynamics. Different
from the common expression of the MRTA problem, the expression in this section belongs to
the continuous optimization problem. Besides, after introducing the replicator dynamics to the
proposed model, the expression includes utility function and stable function. To solve the
proposed model, the EO algorithm is selected and illustrated in Section 4. Simulation results
and discussions about the applicability of the proposed model, the impacts of the total tasks
and the effectiveness of the EO algorithm are shown detailedly in Section 5. At last, conclusions
are drawn in Section 6.

After adding contributions to the structural flow, the structure of the article is clearer
and the research significance of the manuscript can also be effectively restated.
Thanks
again for the suggestions.

Comment No. 8:

The contributions of this article should be improved in comparison with the existing
literature. Compared with existing results, the advantages of the paper should be
further highlighted and the occurred difficulties of conducted topic can be explained.

Response to comment No. 8:

Many thanks for your helpful suggestions. As for the comparison with the existing
literature, we searched a paper with a similar aim of our paper. The citation of it is
shown in:

Page 2, Line 92-97. Therefore, the MRTA problem for the AUVs needs to get the
reliably pre-set task allocation values. There exsit two main approaches to solve MRTA
problems, which are market-based approaches and optimization-based approaches. For
the hash environment, several allocation methods are designed to overcome it. For
example, the Location-Aided Task Allocation Framework (LAAF) task allocation method is specially designed f to balance the objectives and the individual constraints of the AUVs [18]. Similar to the LAAF method, inspired
As for the comparison with the existing results,
because there is a big difference
between the proposed MRTA method and the common MRTA method, we are sorry
for that there is no simulation results similar to ours in this field. However, by
comparing with the other similar researches, we figured out several difficulties of the
conducted topic, which are shown in:

Page 17, Line 567-570.
Compared with the existing literatures, the difficulties of this
research are as follows. Firstly, there are no powerful underwater robots designed for
cleaning marine plastics in the market. Secondly, to make this research practical, there
will be a long way for us to study.

After this kind of modifications, the paper gets more theoretical support. Thanks again
for your helpful suggestions.

Comment No. 9:

In the simulation part, more design parameters are recommended. A detailed discussion
of the figures is helpful to illustrate the results. It is better to explain how the values of
the parameters in the proposed method are adjusted?

Response to comment No. 9:

Thanks for your helpful suggestions. We added the reasons why the values of the
parameters are chosen. The modifications are shown in:

Page 12, Line 442-444.
AUV is known to be expensive, bulky and complicated.
Therefore, three AUVs including 1 S-AUV, 1 M-AUV, and 1 L-AUV are enough for
a small-scale system.

Page 12, Line 453-454.
Initially, considering the common loads of the AUV and the
estimated cleaning need of the marine plastics,

After added the reasons, the parameter setting becomes more convincing. Thanks again
for your careful review.

Comment No. 10:

To confirm the effectiveness and practicality of the proposed approach, an experiment
is strongly suggested since the theoretical approach of this study seems ordinary.

Response to comment No. 10:

Thanks for your comments.
However, existing AUVs cannot achieve the function we
set in the optimization problem. Therefore, experiments cant be done effectively now.
In the future, we may design a AUV with related functions on the basis of this paper.
The future research direction can also be the experimental verification of this paper.
Thanks again for your careful review
.

Comment No. 11:

The readability and presentation of the work should be improved. Some grammar issues
and typos need to be corrected.

Response to comment No. 11:

Thanks for your reminder. We have checked the paper and corrected many grammar
issues and typos. Thanks again for your helpful comments and suggestions.

Round 2

Reviewer 2 Report

Thank you for the author’s response. The authors have made substantial revisions that have covered all my concerns. Most of the recommendations I made were addressed by the authors, therefore I believe that the paper is now ready for publication.